# In Vitro and In Situ Characterization of the Intestinal Absorption of Capilliposide B and Capilliposide C from *Lysimachia capillipes* Hemsl

**DOI:** 10.3390/molecules24071227

**Published:** 2019-03-28

**Authors:** Xu Zhang, Xiao Cheng, Yali Wu, Di Feng, Yifan Qian, Liping Chen, Bo Yang, Mancang Gu

**Affiliations:** 1College of Pharmacy, Zhejiang Chinese Medical University, Hangzhou 311402, China; zhangxu@zcmu.edu.cn (X.Z.); wuyali@zcmu.edu.cn (Y.W.); fengdi@zcmu.edu.cn (D.F.); qianyifan@zcmu.edu.cn (Y.Q.); LipingChen@zcmu.edu.cn (L.C.); Yangbo@zcmu.edu.cn (B.Y.); 2Huzhou Institute for Food and Drug Control, Huzhou 313000, China; cheng_xiao1981@163.com

**Keywords:** capilliposide B, capilliposide C, intestinal absorption, caco-2 cell, single-pass intestinal perfusion, liquid chromatography-tandem mass spectrometry

## Abstract

The goal of this investigation was to determine the processes and mechanism of intestinal absorption for capilliposide B (CAPB) and capilliposide C (CAPC) from the Chinese herb, *Lysimachia capillipes* Hemsl. An analysis of basic parameters, such as drug concentrations, time, and behavior in different intestinal segments was analyzed by liquid chromatography-tandem mass spectrometry (LC-MS). The susceptibility of CAPB and CAPC to various inhibitors such as P-glycoprotein (P-gp) inhibitor (verapamil); multidrug resistance-associated protein 2 (MRP2) inhibitor (indomethacin); cytochrome P450 protein 3A4 (CYP3A4) inhibitor (ketoconazole); and the co-inhibitor of P-gp, MRP2 and CYP3A4 (cyclosporine A) were assessed using both caco-2 cell monolayer and single-pass intestinal perfusion (SPIP) models. As a result, CAPB and CAPC are both poorly absorbed in the intestines and exhibited segment-dependent permeability. The intestinal permeability of CAPB and CAPC were significantly increased by the co-treatment of verapamil, indomethacin. In addition, the intestinal permeability of CAPB was also enhanced by ketoconazole and cyclosporine A. It can be concluded that the intestinal absorption mechanisms of CAPB and CAPC involve processes such as facilitated passive diffusion, efflux transporters, and enzyme-mediated metabolism. Both CAPB and CAPC are suggested to be substrates of P-gp and MRP2. However, CAPB may interact with the CYP3A4 system.

## 1. Introduction

The oral delivery route for the therapeutic administration of drugs remains one of the most desirable and important routes in drug delivery. With a combination of increased patient compliance, safety and ease of administration, orally-delivered drugs often offer greater clinical efficacy than other options, particularly in minimizing potential infections [1]. However, many therapeutics are unable to utilize this sought-after delivery route due to the poor solubility and instability of many compounds in gastrointestinal fluids. Furthermore, properties such as rapid metabolic elimination, low intestinal permeability, and efflux by protein transporters are major obstacles to the oral delivery of various compounds [2]. In recent years, there have been many attempts to overcome these hurdles described by the aforementioned limitations on oral administration.

The efflux transporters, such as the adenosine triphosphate-binding cassette transporters (ABC transporters), play a critical role in the absorption and distribution of drugs in the intestinal organs. These transporters are important factors dictating the bioavailability of the administered drug in addition to the enzymes responsible for drug metabolism [3,4]. ABC transporters such as, P-glycoprotein (P-gp), multidrug resistance-associated proteins (MRPs) and breast cancer resistance protein (BCRP) are capable of excreting drugs in the cellular compartment into the interstitial compartment, decreasing bioavailability [5]. Many active components of traditional Chinese medicine (TCM), such as saponins (akebia saponin D, ginsenoside Rh2, araloside A, the total saponins of Mao-Dong-Qing), flavones (apigenin, epimedins) and alkaloids (reserpine, vincristine, vinblastine) [6,7,8,9,10,11,12] have been shown to be substrates of efflux transporters, which negatively affect the absorption of drugs in the intestine and result in poor bioavailability.

Aside from ABC transporters, drug metabolism enzymes are also critical factors in determining the bioavailability of drugs, among which the cytochrome P450 enzyme (CYP450) plays a major role [13]. The CYP3A4 subset of CYP450 accounts for 30% of the total CYP450 enzyme system [14,15] and is responsible for the bio-transformation of several drugs with a key role in pharmacological effects.

Several permeability models have been employed to understand mechanisms of intestinal absorption, including an in vitro epithelial cell monolayer model such as caco-2 monolayer model and MDCK-MDRI epithelial cell monolayer model [16,17], in vitro everted intestinal sac [18], in vivo or in situ intestinal perfusion models [19], and in vivo pharmacokinetic models. Among these permeability models, the caco-2 cell monolayer model and in situ intestinal perfusion (SPIP) technique are considered as the gold standard in intestinal permeability research and have been wildly utilized to predict intestinal permeability [20,21]. Caco-2 cells express active transport systems, such as P-gp and MRPs [5,22]. However, because of its low expression levels of P450 enzymes and the absence of a protective mucus layer, the cell lines cannot be used for studying the interaction of transporters with underlying metabolic actions.

*Lysimachia capillipes* Hemsl, a Chinese herb and medicinal plant, is widely used as a remedy for the treatment of colds and arthritis as observed for Cannabis sativa extracts [23,24]. Recently, pharmacological investigations revealed that capilliposide B (CAPB) and capilliposide C (CAPC), the major components of saponins isolated from the Lysimachia genus [25], exhibit an inhibitory effect on cell proliferation in various cancers, such as esophageal cancer [26], human non-small cell lung cancer [27], prostate cancer [28], and nasopharyngeal cancer [29]. Although CAPB and CAPC possess anticancer activity, their poor intestinal absorption obstructs further applications. This poor performance may be attributed to two key factors. On one hand, systemic exposure of CAPB and CAPC was low with significant variation among individuals after oral administration. This is likely a result of its extensive biotransformation in the gastrointestinal tract [30]. On the other hand, the low intestinal permeability of CAPB and CAPC may also serve as a critical barrier, resulting in poor bioavailability and low exposure in tissues after oral administration [31]. However, the specific mechanisms which affect the permeability of CAPB and CAPC through the intestinal epithelium are unknown. Thus, a systemic study is required to investigate the intestinal absorption of CAPB and CAPC.

Therefore, the primary aim of this study was to investigate the process and mechanisms by which CAPB and CAPC are absorbed by intestinal cells using caco-2 cell monolayer and in situ intestinal perfusion models. The effect of parameters such as drug concentration, transport time and behavior in different intestinal segments were analyzed. Furthermore, the susceptibility of CAPB and CAPC absorption to treatment with various inhibitors, such as P-gp inhibitor (verapamil), MRP2 inhibitor (indomethacin); CYP3A4 inhibitor (ketoconazole); and the co-inhibitor of P-gp, MRP2 and CYP3A4 (cyclosporine A) were also assessed.

## 2. Results

### 2.1. Liquid Chromatography-Tandem Mass Spectrometry Analysis Method Validation

The concentration of CAPB and CAPC across caco-2 cell monolayer was detected by liquid chromatography-tandem mass spectrometry (LC-MSn). The total ion chromatogram and product ion mass spectra of CAPB, CAPC and dioscin (IS) are shown in Figure 1, while the standard curves of CAPB and CAPC over the concentrate range of 1.0–5000 ng/mL are shown in Appendix A. The equations of the regression line were y = 40.84x − 8.1502 (r^2^ = 0.998) for CAPB (over 1–250 ng/mL), y = 64.736x − 147.9 (r^2^ = 0.998) for CAPB (over 250–5000 ng/mL), y = 38.961x − 11.532 (r^2^ = 0.987) for CAPC (over 1–250 ng/mL), and y = 68.756x − 104.59 (r^2^ = 0.994) for CAPC (1–5000 ng/mL). Extraction recoveries at concentrations of 10, 150 and 3000 ng/mL were determined to be 91.6%, 98.41% and 98.44% for CAPB; and 82.05%, 96.65% and 81.65% for CAPC, respectively. The matrix effect of CAPB and CAPC was between 1.01 and 1.11 (RSD < 3.3%) as evaluated internally by a standard-normalized matrix factor. Intra-day and Inter-day variations were both less than 4%.

### 2.2. The Characterization of Caco-2 Cell Monolayer

The integrity of the monolayer was evaluated by measuring the trans-epithelial electrical resistance (TEER) and phenol red permeability studies. In our data, the monolayer displayed a TEER values >300 Ωcm^2^ andthe apparent permeability (P_app_, cm/s) values of Phenol <10^5^ cm/s after growing for 20 days. These results indicate that caco-2 cell monolayer may be used for permeability studies. In addition, cell viability was verified by MTT assay. As shown in Appendix A, CAPB and CAPC less than 40 μg/mL did not inhibit cell growth significantly. However, a concentration of CAPB and CAPC over 40 μg/mL resulted in significantly decreased cell viability. Hence, 10, 20 and 40 μg/mL of CAPB and CAPC was selected as the testing concentrations for the drug transport study.

### 2.3. The Characterization of the Intestinal Permeability Features of CAPB and CAPC In a Caco-2 Cell monolayer

A caco-2 cell monolayer model was used to explore the intestinal permeability features of CAPB and CAPC. Firstly, the P_app_ values were measured at different drug concentrations (10, 20, 40 μg/mL), and the P_app_ of CAPB and CAPC were all found to be less than 2 × 10^−6^ cm/s, as shown in Figure 2A,B. Our data also showed that P_app_ values increased as the CAPB and CAPC concentration increased to high and medium concentrations. Both CAPB and CAPC displayed a significant increase in permeability compared to lower concentrations (*p* < 0.05). However, no significant difference was found between high concentrations and medium concentrations of CAPB and CAPC (*p* > 0.05). Furthermore, the P_app_ values obtained after incubation with CAPB and CAPC for 45, 60 and 90 min across caco-2 cell monolayer in the AP-BL and BL-AP direction is presented in Figure 2C. It was found that P_app_ values showed an upward trend over time. As shown in Figure 2D, the P_app_ (BA)/P_app_ (AB) values (efflux ratio values, ER values) of CAPB and CAPC at different initial drug concentrations (10, 20, 40 μg/mL) were more than 1.0 and less than 1.5.

### 2.4. The Role of P-gp, MRP2 and CYP3A4 on CAPB and CAPC Transport Across Caco-2 Cell Monolayer

It was also speculated that the efflux transporter and metabolism enzyme plays an important role in the permeability of CAPB and CAPC across caco-2 cell monolayer as shown in Figure 3. Compared with treatment using CAPs alone, for the co-treatment with either P-gp inhibitor (verapamil); MRP2 inhibitor (indomethacin); or the co-inhibitor of P-gp, MRP2 and CYP3A4 (cyclosporine A), the P_app_ values of CAPB increased about 25 times (*p* < 0.01), 11 times (*p* < 0.05) and 10 times (*p* < 0.05), respectively, on the AP-BL side and around 30 times (*p* < 0.01), 10 times (*p* < 0.05) and 10 times (*p* < 0.05), respectively on the BL-AP side. The P_app_ values of CAPC on the AP-BL side were also significantly increased up to 11 times (*p* < 0.05) and 16 times (*p* < 0.01) in the presence of verapamil and indomethacin, respectively. However, in the presence of cyclosporin A, the permeability of CAPC showed only a small increase (*p* > 0.05), while CAPB showed a 10-fold increase (*p* < 0.05). Therefore, the in vitro transport data indicates that CAPB and CAPC may be the substrate of the efflux protein P-gp and MRP2; moreover CAPB may also be affected by CYP3A4.

### 2.5. The Characterization of the Intestinal Permeability of CAPB and CAPC In Rats

The SPIP model was used to further explore the intestinal permeability features of CAPB and CAPC in rats. Firstly, we determined the stability of CAPB and CAPC in the Krebsringer buffer (K-R buffer) across different pH values. As shown in Figure 4A, CAPB and CAPC were more stable at pH 5.0 and pH 6.55 than pH 7.43. In other words, CAPB and CAPC were stable in a weakly acidic environment. Because the pH of intestinal juice was close to 6.55, we used a K-R buffer at pH 6.55 as the perfusion solution.

The effective permeability (P_eff_, cm/s) and absorption rate constants (ka, s^−1^) values of CAPB and CAPC were measured at different drug concentrations (20, 50 and 80 μg/mL). As shown in Figure 4B, the P_eff_ and Ka values of CAPB showed a slight decline between 20 and 50 μg/mL. However, a significant increase was seen at 80 μg/mL of drug concentration. However, the P_eff_ and Ka values of CAPC showed a slight increase between 20 to 50 μg/mL followed by a significant decrease (*p* < 0.05) as drug concentration increased.

Subsequently, the P_eff_ and Ka values of CAPB and CAPC in the different intestinal segments including duodenum, jejunum and ileum were assessed. As shown in Figure 4C, the order of P_eff_ and Ka values of CAPB in three different intestinal segments was duodenum > jejunum > ileum. The order of CAPC followed the same pattern as CAPB. The permeability in the duodenum was significantly greater than that in jejunum and ileum, respectively (*p* < 0.05). Our data demonstrated that CAPB and CAPC may exhibit segmental-dependent permeability and were best absorbed in the duodenum.

### 2.6. The Role of P-gp, MRP2, and CYP3A4 on Intestinal Permeability of CAPB and CAPC

To further confirm the role of P-gp, MRP2 and CYP3A4 on the intestinal permeability of CAPB and CAPC, the P_eff_ and Ka values were measured in the presence of verapamil, indomethacin, ketoconazole, and cyclosporine A. As the findings in Figure 5 indicate, The P-gp inhibitor, verapamil, resulted in a 1.5-fold increase (*p* < 0.05) on the intestinal transport of CAPB and CAPC. In the presence of indomethacin, the Ka and P_eff_ values increased significantly (*p* < 0.05). The P_eff_ value of CAPB showed an even more significant increase at 4.2 fold while the P_eff_ of CAPC only increased 2 fold. It is worth noting that the P_eff_ values of CAPB and CAPC are equal in the absence of an inhibitor. Therefore it is indicated that the affinity between CAPB and MRP2 is stronger than that of CAPC.

In the presence of the CYP3A4 inhibitor, ketoconazole, CAPB showed a two-fold higher permeability than CAPB alone (*p* < 0.05). However, CAPC exhibited minimal alteration of permeability in the presence of ketoconazole (*p* > 0.05). The substrate of CYP3A4 protein is therefore elucidated to be CAPC over CAPB. In the presence of cyclosporin A, a co-inhibitor of P-gp, MRP2, and CYP3A4, the permeability of CAPC increased slightly (*p* > 0.05), while CAPB increased around two-fold (*p* < 0.05).

## 3. Discussion

CAPB and CAPC, two major compounds of *L. capillipes* Hemsl, exhibit significant cytotoxicity against many human cancer cell lines, including prostate cancer cell PC3 and DU145, along with nasopharyngeal cancer CNE-2 cells; ovarian cancer SK-OV-3 and A2780 subtypes; and lung cancer PC-9, A549, H1299, and H460 cells [25,26,27,28,29,30]. Moreover, CAPB and CAPC were found to inhibit tumor growth without inducing significant toxicity to hepatic or renal tissues in a mouse model [26,27,32]. Previously, studies have evaluated the pharmacokinetics, tissue distribution, and excretion of CAPB and CAPC systematically [31,33,34]. Both compounds showed poor bioavailability and low exposure in tissues after oral administration [31]. However, the intestinal absorption mechanics and underline transport systems of CAPB and CPAC were still unclear. In this study, we revealed that both CAPB and CAPC showed low permeability across intestinal epithelial cells. However, the intestinal absorption of CAPB and CAPC may involve facilitated passive diffusion, and may be affected by efflux transporters and metabolic enzymes.

In a caco-2 cell model, our data showed that the P_app_ of CAPB and CAPC were both less than 2 × 10^−6^ cm/s, which represents a low permeability of compounds in vitro. The amount of CAPB and CAPC across the cell monolayer accumulated as concentration and time increased. Meanwhile, the ER values of CAPB and CAPC were between 1.00 and 1.50, indicating that facilitated diffusion and efflux mechanisms may be involved in the intestinal epithelium transportation of CAPB and CAPC [22,35]. On the other hand, P_eff_ values of CAPB and CAPC were both much less than 0.2 × 10^−4^ cm/s in a single-pass intestinal perfusion model, which confirmed the in vitro transportation results. Interestingly, we found that the P_eff_ of CAPB and CAPC fluctuated only slightly from 20 to 50 μg/mL but were drastically restrained at 80 μg/mL in an in situ intestinal infusion model. A possible explanation was that the facilitated passive diffusion may involve intestinal transport, similar to the transport mechanism of sodium taurocholate [36]. However, we did not assess the transportation features of CAPB and CAPC at 80 μg/mL in the caco-2 monolayers model, as the drug showed cytotoxicity when the concentration of CAPB and CAPC was above 40 μg/mL. Because the P_app_ of CAPB and CAPC increased linearly from 10μg/mL to 40 μg/mL in the caco-2 monolayer model, we wanted to further characterize the intestinal permeability of compounds at higher concentrations in the SPIP model. In addition, this data showed that P_eff_ and Ka values of CAPB and CAPC in the duodenum were significantly higher than that in the jejunum and ileum. This may be related to pKa of the drug, the degree of dissociation, the pH in the four intestinal segments, the relative abundance of microvilli and villi, and the distribution of efflux transporters and uptake transporters [37].

The P-gp and MRP2 proteins are two major efflux transporters which affect the absorption of drugs in the intestines [38,39] and are shown to be highly expressed in intestinal epithelium as well as on the membranes of caco-2 cells [4]. Numerous studies have shown that active components of traditional Chinese medicine are the substrates of P-gp and MRP2 proteins, which are likely explanations for the reduced intestinal absorption of oral ginsenoside Rh2, a member of the saponins [40]. As examples, akebia saponin D demonstrates poor intestinal absorption as a result of MRPs in the intestine [41]. Ginsenoside Rh2 [42,43] and araloside A [22] have poor intestinal absorption, because they are both substrates of P-gp.

This in vitro transport data and in situ intestinal infusion data showed that the inhibition of P-gp and MRP2 activity could significantly enhance the permeability of CAPB and CAPC across intestinal epithelia. These results indicated that CAPB and CAPC, also members of the saponins family, may be substrates for the efflux protein P-gp and MRP2. The intestinal permeability of CAPB and CAPC may, at least partly, be limited by P-gp and MRP2.

In addition to the ABC transport protein, metabolic enzymes play a critical role in the intestinal absorption of drugs [44]. For example, CYP3A4 was shown to also influence ginsenoside Rh2 intestinal absorption as it was the predominant enzyme responsible for the oxidation of ginsenoside Rh2 [43,45]. The best example perhaps, is with paclitaxel, whose poor bioavailability is caused by a combination of poor water solubility, P-gp efflux, and CYP3A4 metabolism. In an excretion study of CAPB, it was proven that CAPB experienced extensive metabolism prior to excretion [34]. Likewise, CAPC also demonstrated extensive metabolism in rat intestinal microflora and a strong anticancer activity [46]. Prior studies also systematically characterized 19 metabolites of CAPB and CAPC in mice and proposed a major metabolic pathway (deglycosylation and esterolysis) following oral dosing [32]. However, caco-2 cells do not always express the appropriate amount of metabolic enzymes, such as P450 enzymes, which can affect the uptake of certain drugs that are transported through metabolic-specific pathways [4,47]. Because of its low expression levels of P450 enzymes, the role of CYP3A4 was investigated primarily by adding the co-inhibitor of P-gp, MRP2 and CYP3A4 (cyclosporin A) in the caco-2 cells and SPIP model. Furthermore, the inhibitor of CYP3A4 (ketoconazole) was added to the SPIP model only for the reasons mentioned above. Our in vitro transport data and in situ intestinal infusion data showed that the inhibition of CYP3A4 could significantly enhance the permeability of CAPB across the intestinal epithelia. It was found that CAPB may be the substrate of CYP3A4, but CAPC may not.

While the results are promising, the current study still suffers from certain limitations. The intestinal absorption of drugs was affected by numerous factors, such as transporters, and intestinal microflora [37]. However, the scope of this study sought to focus on the permeability features of CAPB and CAPC, along with the major transporters and metabolic enzymes in intestinal epithelium including P-gp, MRP2 and CYP3A4 [5,13]. While previous studies have already evaluated the pharmacokinetics, distribution, intestinal metabolism, and excretion of CAPB and CAPC [31,32,33,34], the absorption characteristics of CAPB and CAPC in the intestinal tract were largely unknown. Additionally, classical inhibitors were used to test the possible drug transport mechanisms instead of a knockout cell or mouse model. This was chosen to make the study more straightforward. In future studies, our group seeks to focus on the effects of intestinal microflora, along with other transporters and metabolic enzymes on the intestinal absorption mechanisms of CAPB and CAPC. A gene-editing protocol may also be employed to further investigate the transport mechanisms of CAPB and CAPC.

## 4. Materials and Methods

### 4.1. Materials

Capilliposide B (C_58_H_96_O_24_, CAPB), capilliposide C (C_57_H_94_O_24_, CAPC) and *Lysimachia capillipes* Hemsl API (more than 70% total of CAPB and CAPC) were obtained from Professor Tian Jingkui, College of Biomedical Engineering and Instrument Science, Zhejiang University (Zhejiang, China). Dioscin (Internal standard, IS, purity ≥ 98%) and Verapamil Hydrochloride were obtained from National Institutes for Food and Drug Control (Beijing, China). Phenol Red, Ketoconazole, Novobiocin and Cyclosporin (purity ≥ 98%) were purchased from Shanghai Yuanye Biotechnology Co., Ltd. (Shanghai, China). The caco-2 human cell line was obtained from the Shanghai Cell Bank of The Chinese Academy of Sciences. HPLC-grade methanol and acetonitrile and other chemicals were of analytical grade.

### 4.2. Liquid Chromatography-Tandem Mass Spectrometry (LC-MS) Analysis

The liquid chromatography system used in this study was an Agilent Technologies model 1290 Infinity (Agilent Technologies, Santa Clara, CA, USA). Separations were carried out using a UItimate XB-C18 column (250 × 4.6 mm, 5 µm, Phenomenex, Toran, CA, USA) at 40 °C. The mobile phase was composed of 0.3% formic acid in water as mobile phase A (MA) and acetonitrile as mobile phase B (MB) using a gradient elution of 49% MB (0–10 min) and 49% to 90% MB from 10–25 min. Separation was carried out at a flow rate of 1.0 mL/min. The sample injection volume was 10 μL.

An Agilent 6460 Triple Quad mass separometer (Agilent Technologies) equipped with a Turboionspray source (TIS) was operated in the positive ionization mode with multiple reaction monitoring (MRM) for LC-MS analysis. The MS parameter was optimized as follows: TIS temperature, 600 °C; ionspray voltage, −4500 V; curtain gas, nitrogen, 30 psi; nebulizing gas, 50 psi; declustering potential, 135 V for CAPB, 135 V for CAPC and 90 V for dioscin (IS); entrance potential, 10 V; collision energy 10 eV for CAPB, CAPC and dioscin (IS); collision cell exit potential, 15 V. The following MRM transition was used: *m*/*z* 1197→1060.6 for CAPB, 1182.8→1022.0 for CAPC, 869.5→725.6 for dioscin.

### 4.3. Caco-2 Cell Culture

Caco-2 cells were routinely cultured in high glucose Dulbecco’s modified Eagle’s medium (DMEM) containing 10% fetal bovine serum (FBS), 1% nonessential amino acids, and 1% penicillin/streptomycin. Cells were kept at 37 °C in a 90% relative humidity atmosphere containing 5% CO_2_. Cells were seeded onto transwells purchased from Corning Costar Co. (New York, NY, USA) on 12-well plates for transport studies and uptake studies at 5 × 10^4^ cells per insert. Cells were grown for 20 days. The integrity of monolayer was evaluated by measuring the trans-epithelial electrical resistance (TEER) and by phenol red permeability studies.

The effects of CAPs on caco-2 cell viability were checked by 3-(4,5-dimethyl-2-thiazolyl)-2,5-diphenyl-2-H-tetrazolium bromide (MTT) colorimetric assay [12], which is adapted to analyze cell proliferation and drug cytotoxicity. Cells with decreased viability are considered to be less metabolically active and hence will reduce less MTT. In brief, caco-2 cells were seeded at 4 × 10^3^ cells/well into 96-well culture plates and were incubated at 37 °C for 24 h before the assay. The cells were treated with different concentrations (0, 10, 20, 40, 80, 120, 160 and 240 μg/mL) of CAPs. After 24 h of incubation, 20 µL of MTT (5 mg/mL) was added to each well and the cells were incubated for another 4 h. After removing the culture medium, 100 µL of dimethyl sulfoxide was added to dissolve the contents in the plate. Then, the absorbance was measured at 570 and 630 nm (reference wavelength) using a microplate reader (Multiskan MK3; Thermo Fisher Scientific, Waltham, MA, USA).

### 4.4. Permeability Studies Using Caco-2 Cells

DMEM was removed and the monolayer was washed with Hank’s Balanced Salt Solution (HBSS). The blank HBSS was replaced by 0.4 mL of HBSS containing 40, 20 and 10 μg/mL of drug on the apical (AP) side and 2.1 mL of HBSS on the basolateral (BL) side. In the AP-BL direction studies, 0.4 mL of HBSS containing 40 μg/mL drug and P-gp inhibitor (53 μg/mL verapamil); MRP2 inhibitor (17.9 μg/mL indomethacin); or the co-inhibitor of P-gp, MRP2 and CYP3A4 (10 μg/mL cyclosporin A) were added to the AP side; 2.1 mL of HBSS was added to the BL side. In the BL-AP direction studies, 0.4 mL of HBSS containing 40 μg/mL drug and different inhibitors were added to the BL side, and 2.1 mL of HBSS was added to AP side. Samples (0.4 mL) were taken from the BL or AP side after 45, 60 and 90 min incubation at 37 °C. An additional 0.4 mL of HBSS was added at the same time. The concentrations of samples were analyzed by LC-MS.

### 4.5. In Situ Single-Pass Intestinal Perfusion (SPIP) Studies In Rats

Male Sprague–Dawley (SD) rats (280 ± 30 g) were supplied by the Laboratory Animal Center, Zhejiang Chinese Medical University (Zhejiang, China). All animals were fasted overnight (12–18 h) with free access to water before experiments. All experiments were performed in accordance with the guidelines for the care and use of animals as established by the Laboratory Animal Centre, Zhejiang Chinese Medical University (Zhejiang, China) (approval number: ACXK20150016).

The perfusion experiment was performed as previously reported. In brief, rats were fasted overnight but permitted to drink water freely the day before the experiments. After being anesthetized, the rats were placed on the surface of a thermostatic device and maintained at 37 °C. An incision of approximately 3 cm was made along the midline of the abdominal cavity to expose the contents of the abdomen. Perfusate entered the duodenum (1 cm below pylorus) and exited from the jejunum (15 cm from the pylorus). Then the incision was made at both sides of the segment. The intestinal contents were rinsed with saline preheated at 37 °C, drained with air, and connected to the perfusion system with the catheter.

One-hundred millitlers of pre-prepared perfusate was taken and preheated at 37 °C. The perfusion was started at a circulation speed of 1.0 mL/min for 10 min. The flow rate was subsequently increased to 0.2 mL/min for 30 min to ensure steady-state conditions. Samples were collected in glass tubes (2 mL/per) at 10-min intervals for 90 min. The samples were filtered through a membrane filter (0.45 μm) and the concentration of CAPB, CAPC and phenol red were measured by LC-MS. The rest of the samples were stored at −20 °C for further studies.

### 4.6. Data Analysis

The apparent permeability (P_app_, cm/s) across caco-2 cell monolayer was calculated from the linear plot of drugs accumulated in the receiver side versus time using Equation (1):(1)$$P_{\mathrm{app}}=\left(\frac{1}{C_{0}A}\right)\left(\frac{\mathrm{dQ}}{\mathrm{dt}}\right)$$
where dQ/dt represents the steady-state flux of the drug on the receiver (serosal in the case of AP-BL studies or mucosal in the case of BL-AP studies) side, C_0_ is the initial concentration of the drug in the donor side, and A is the monolayer growth surface area (4.67 cm^2^). Linear regression was carried out to obtain the steady-state appearance rate of the drug on the receiver side.

Effective permeability (P_eff_) and absorption rate constants (Ka) were calculated using the following Equations (2), (3) and (4), respectively
(2)$$\frac{{C^{\prime }}_{\mathrm{out}}}{{C^{\prime }}_{\mathrm{in}}}=\frac{C_{\mathrm{out}}}{C_{\mathrm{in}}}\times \frac{C_{\mathrm{in},\mathrm{phenolred}}}{C_{\mathrm{out},\mathrm{phenolred}}},$$
(3)$$P_{\mathrm{eff}}(m/s)=\frac{{-\mathrm{Qln}(C^{\prime }}_{\mathrm{out}}/{C^{\prime }}_{\mathrm{in}})}{2\pi \mathrm{RL}},$$
(4)$$\mathrm{Ka}=Q(1-\frac{C^{\prime }{}_{\mathrm{out}}}{C^{\prime }{}_{\mathrm{in}}}{)\pi R}^{2}L,$$
where C_in_ phenol red and C_out_ phenol red is equal to the concentrations of phenol red in the inlet and outlet samples, respectively; C′_out_/C′_in_ is the ratio of the outlet and inlet concentration of the tested drug that has been adjusted for water transport, Q is the perfusion buffer flow rate (0.2 and 0.1 mL/min for rats and mice, respectively), R is the radius of the intestinal segment (set to 0.2 and 0.1 cm for rats and mice, respectively), and L is the length of the intestinal segment.

### 4.7. Statistical Analysis

All experiments were performed in triplicate (minimum) and results were expressed as mean values ± standard deviation (SD). Statistical comparisons were performed by Student’s t-tests or one-way analysis of variance (ANOVA) using the SPSS version 22 software. Comparisons between two groups were analyzed using Student’s t-tests. When a *p*-value was smaller than 0.05, it was considered statistically significant. All data were represented for at least three independent experiments.

## 5. Conclusions

The present study has revealed that CAPB and CAPC are poorly absorbed in the intestines and likely exhibited segmental-dependent permeability; it can also be found that the intestinal absorption mechanism of CAPB and CAPC may involve facilitated passive diffusion associated with the efflux transporters P-gp and MRP2, along with the metabolic enzyme CYP3A4. As a whole, CAPB may be the substrate of the P-gp, MRP2 and CYP3A4, while CAPC may be the substrate of the P-gp and MRP2, but not of the CYP3A4. In conjunction with results from previous studies along the direction of CAPB and CAPC, these results provide updated information concerning the intestinal absorption process and the possible mechanism of these two compounds.

## Figures and Tables

**Figure 1 molecules-24-01227-f001:**
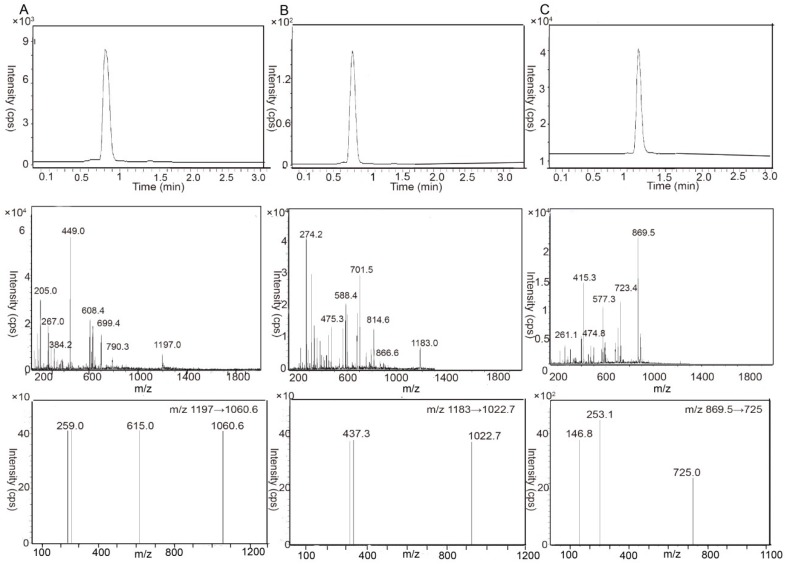
Total ion chromatogram (top panel), product ion mass spectra (middle panel) and the multiple reaction monitoring (MRM) transitions of the deprotonated molecular ions mass spectrogram (bottom panel) of capilliposide B (CAPB, **A**), capilliposide C (CAPC, **B**) and dioscin (IS, **C**). The chromatograms monitoring of CAPB was at *m*/*z* 1197.0→1060.6, CAPC at *m*/*z* 1183.0→1022.7 and IS at *m*/*z* 869.5→725.0.

**Figure 2 molecules-24-01227-f002:**
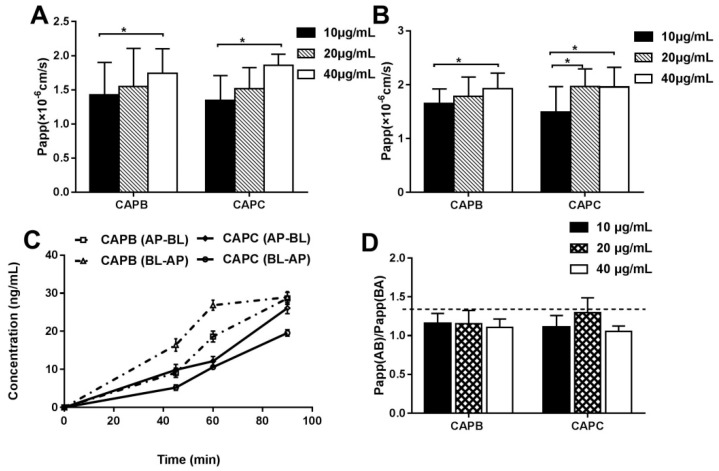
Bidirectional transport studies of capilliposide B (CAPB, **A**), capilliposide C (CAPC, **B**) at different initial drug concentrations (10, 20 and 40 μg/mL). *p* < 0.05 (*), comparison with the 10 μg/mL capilliposides (CAPs) group. The apparent permeability (P_app_, cm/s) values of CAPB and CAPC at 0, 45, 60 and 90 min (**C**). The P_app_ (BA)/P_app_ (AB) values of CAPB and CAPC were at different initial drug concentrations (10, 20 and 40 μg/mL) (**D**). All results are expressed as mean ± S.D. (n = 3).

**Figure 3 molecules-24-01227-f003:**
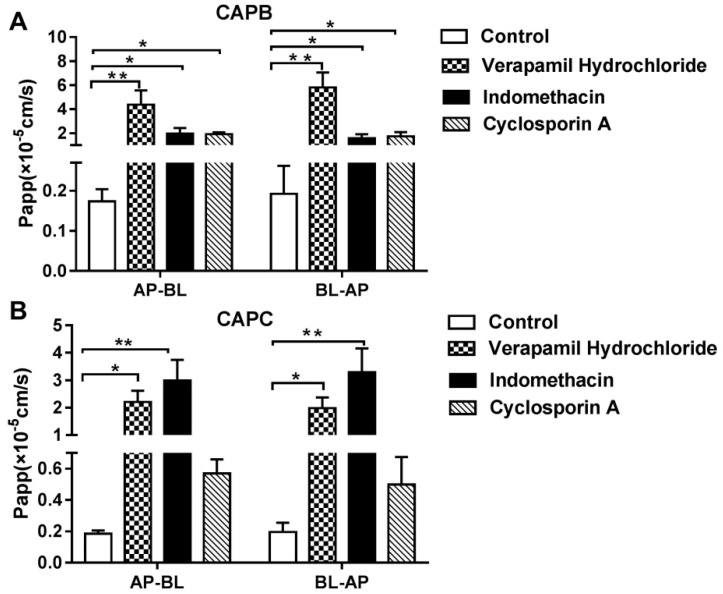
The effect of different factors on the transportation of capilliposide B (CAPB) and capilliposide C (CAPC) across caco-2 cell monolayer. The data are presented as the apparent permeability (P_app_, cm/s). Effect of P-glycoprotein (P-gp) inhibitor (verapamil hydrochioride); multidrug resistance-associated protein 2 (MRP2) inhibitor (indomethacin); and the co-inhibitor of P-gp, MRP2 and cytochrome P450 protein 3A4 (CYP3A4) on caco-2 cell monolayer for CAPB (**A**) and CAPC (**B**). *p* < 0.05 (*), *p* < 0.01 (**), comparison with control. All results are expressed as mean ± S.D. (n = 3).

**Figure 4 molecules-24-01227-f004:**
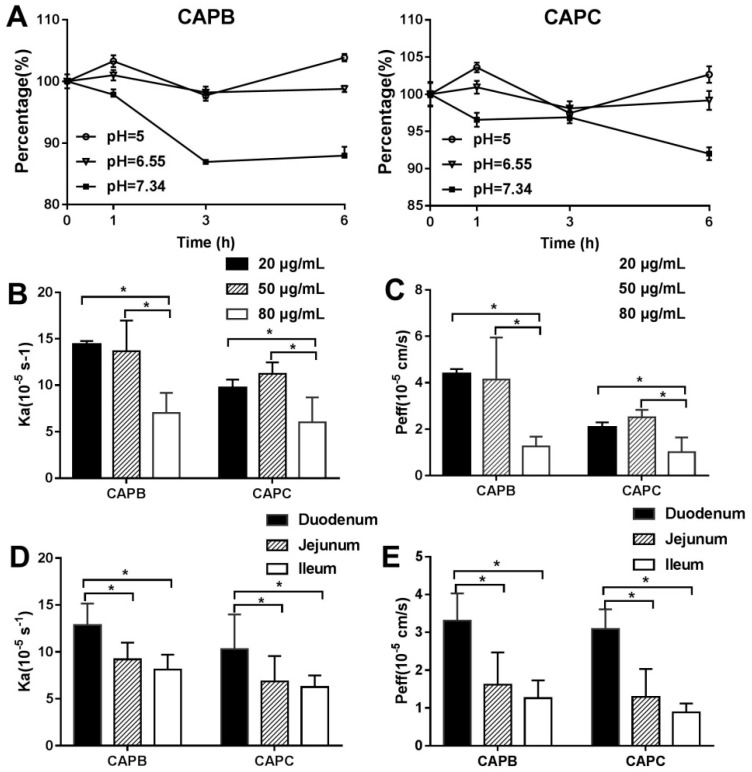
The characterization of the intestinal permeability of capilliposide B (CAPB) and capilliposide C (CAPC) in single-pass intestinal perfusion (SPIP) model. The stability of CAPB and CAPC were measured at different pH values (pH 5, pH 6.55, and pH 7.34, respectively) (**A**). The absorption rate constants (ka, s^−1^) values (**B**) and effective permeability (P_eff_, cm/s) values (**C**) for CAPB and CAPC at different initial drug concentrations (20 μg/mL, 50 μg/mL, and 80 μg/mL, respectively). *p* < 0.05 (*) compared with with the group at 80 μg/mL. The Ka values (**D**) and P_eff_ values (**E**) of CAPB and CAPC obtained from the duodenum, jejunum, and ileum in SPIP models. *p* < 0.05 (*) compared with the duodenum. All the results are expressed as mean ± S.D. (n = 3).

**Figure 5 molecules-24-01227-f005:**
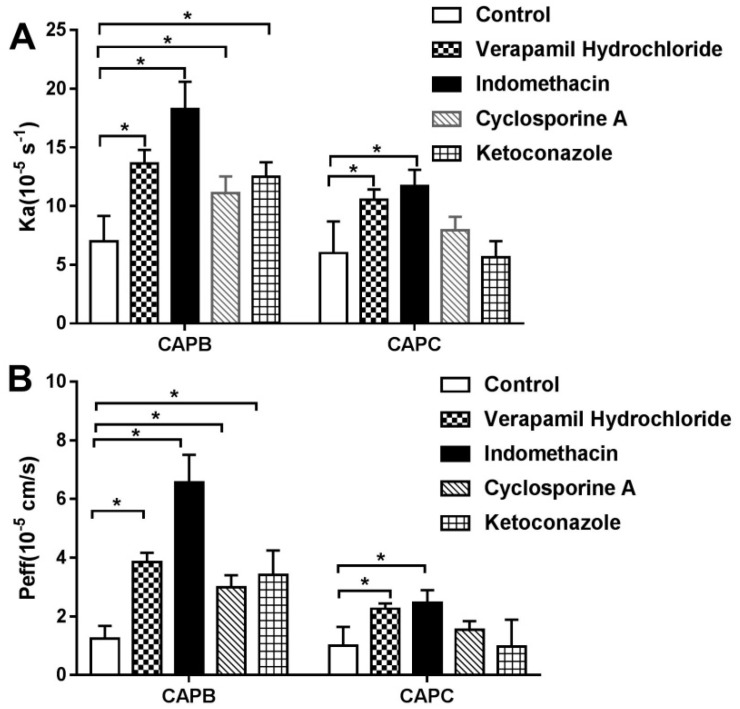
The effect of different factors on intestinal absorption of capilliposide B (CAPB) and capilliposide C (CAPC). The data are presented as absorption rate constants (ka, s^−1^) values (B) and effective permeability (P_eff_, cm/s). P-glycoprotein (P-gp) inhibitor (verapamil hydrochioride); multidrug resistance-associated protein 2 (MRP2) inhibitor (indomethacin); and the co-inhibitor of P-gp, MRP2 and cytochrome P450 protein 3A4 (CYP3A4) (cyclosporine A); and CYP3A4 inhibitor (ketoconazole) on small intestinal absorption of CAPB (**A**) and CAPC (**B**). *p* < 0.05 (*), in comparison with control. All the results are expressed as mean ± S.D. (n = 3).

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
