# Peer review of "In Vitro and In Situ Characterization of the Intestinal Absorption of Capilliposide B and Capilliposide C from Lysimachia capillipes Hemsl"

_molecules, 2019, doi:10.3390/molecules24071227_

Round 1
Reviewer 1 Report
It was a pleasure to review the review "In Vitro and In Situ Characterization of the Intestinal Absorption of Capilliposide B and Capilliposide C from Lysimachia capillipes Hemsl". Zhang and colleagues did a great phyto-pharmacological overview of Capilliposides with a focus on the cytochromes and metabolism. The description of these compounds is very accurate. The findings presented here provide a new incentive to explore the Lysimachia capillipes Hemsl as therapeutic tool for the treatment of different diseases.
However, I have some comments to address to the authors to improve and clarify some parts of this review.
1) In the Introduction chapter, authors should discuss other methods used to study permeability;
2) In the text, I found some typing and/or punctuation errors, a better linguistic and stylistic revision might be necessary (for example: ketocolnazole or ketoconazde);
3) Is MTT assay a valid test for viability?
4) The authors must justify why they did not use a lower dose of 20 ug/ul of CAPB;
The bibliographic collection is accurate, but some more recent references should be added. In particular:
- line 71, Authors should add “as observed for Cannabis sativa estracts” and insert “ Bonini SA et al. Cannabis sativa: A comprehensive ethnopharmacological review of a medicinal plant with a long history. J Ethnopharmacol. 2018”.
Author Response
XuZhang et al. MOLECULES/2019/ 464121
Dr. Derek J. McPhee
Editor-in-chief, Molecules
Reply to the Reviewers
Thank you very much for your interest in our manuscript entitled “In Vitro and In Situ Characterization of the Intestinal Absorption of Capilliposide B and Capilliposide C from Lysimachia capillipes Hemsl.” To aid in the re-review of this manuscript, we have included a point-by-point response to each comment. he reviewers’ comments are italicized and placed in square brackets. In addition, within the revised manuscript, we have used underlined text to highlight changes in response to the reviewers’ comments.
We appreciate the suggestions and comments by the reviewers. As a consequence of the valuable suggestions, we believe that our manuscript has been much improved.
[Reviewer 1: Comments for the Authors...]
[Questions and comments:]
[1. In the Introduction chapter, authors should discuss other methods used to study permeability.]
We appreciate the suggestion. Several permeability models have been employed to understand characterization and mechanisms of intestinal absorption, including in vitro epithelial cell monolayer model such as caco-2 monolayer model and MDCK-MDRI epithelial cell monolayer model, in vitro everted intestinal sac, in vivo or in situ intestinal perfusion models, and in vivo pharmacokinetic models. Accordingly, we have modified the INTRODUCTION section as follows:
“Several permeability models have been employed to understand mechanisms of intestinal absorption, including in vitro epithelial cell monolayer model such as caco-2 monolayer model and MDCK-MDRI epithelial cell monolayer model [16-17], in vitro everted intestinal sac [18], in vivo or in situ intestinal perfusion models [19], and in vivo pharmacokinetic models.”
(INTRODUCTION, page 2)
“16. Wang, X. X.; Liu, G. Y.; Yang, Y. F.; Wu, X. W.; Xu, W.; Yang, X. W., Intestinal Absorption of Triterpenoids and Flavonoids from Glycyrrhizae radix et rhizoma in the Human Caco-2 Monolayer Cell Model. Molecules 2017, 22 (10). doi:10.3390/molecules22101627
17. Chen, Z. L.; Ma, T. T.; Huang, C.; Zhang, L.; Zhong, J.; Han, J. W.; Hu, T. T.; Li, J., Efficiency of transcellular transport and efflux of flavonoids with different glycosidic units from flavonoids of Litsea coreana L. in a MDCK epithelial cell monolayer model. Eur J Pharm Sci 2014, 53, 69-76. doi: 10.1016/j.ejps.2013.12.010
18. Gurunath, S.; Nanjwade, B. K.; Patila, P. A., Enhanced solubility and intestinal absorption of candesartan cilexetil solid dispersions using everted rat intestinal sacs. Saudi Pharm J 2014, 22 (3), 246-257.
19. Xin, L.; Liu, X. H.; Yang, J.; Shen, H. Y.; Ji, G.; Shi, X. F.; Xie, Y., The intestinal absorption properties of flavonoids in Hippophae rhamnoides extracts by an in situ single-pass intestinal perfusion model. J Asian Nat Prod Res 2017, 1-14. doi:10.1080/10286020.2017.1396976”
[2. In the text, I found some typing and/or punctuation errors, a better linguistic and stylistic revision might be necessary (for example: ketocolnazole or ketoconazde.]
Thanks for pointing out the errors in the manuscript. We examined the full text carefully and made a linguistic revision. To address the point raised, we have revised as follows:
Line 23: “cytochrome P450 protein 3A4 (CYP3A4) inhibitor (ketoconazole)”
line 29: “In addition, the intestinal permeability of CAPB was also enhanced by ketoconazole and cyclosporine A.”
Line 89: “CYP3A4 inhibitor (ketoconazole) and the co-inhibitor of P-gp, MRP2, and CYP3A4 (cyclosporine A) were also assessed.”
Line 179: “in the presence of verapamil, indomethacin, ketoconazole and cyclosporine A.”
Line 187: “In the presence of the CYP3A4 inhibitor, ketoconazole,”
Line 189: “in the presence of ketoconazole (p>0.05).”
line 196: “and CYP3A4 inhibitor (ketoconazole) on small intestinal absorption for CAPB (A) and CAPC (B).”
Line 256: “the inhibitor of CYP3A4 (ketoconazole) was added to the SPIP model only for the reasons mentioned above.”
Line: 282: “Phenol Red, Ketoconazole, Novobiocin and Cyclosporin (purity ≥ 98%) were purchased from Shanghai Yuanye Biotechnology Co., Ltd (Shanghai, China).”
Line 383: “The standard curve of CAPB and CAPC across caco-2 cell monolayer was detected by LC-MS.”
Line 385: “XuZhang wrote and substantively revised the manuscript.”
[3. Is MTT assay a valid test for viability?]
We appreciate the question. MTT assay is well known to study cell proliferation and drug cytotoxicity. It monitors metabolic activity of cultured cells by quantitating dark-colored formazan produced by the reduction of the tetrazolium salt MTT by metabolically active cells. Cells with decreased viability are considered to be less metabolically active and hence will reduce less MTT. Accordingly, we have modified the MATERIAL AND METHODS section as follows:
“The effects of CAPs on caco-2 cell viability were checked by 3-(4,5-dimethyl-2-thiazolyl)-2,5-diphenyl-2-H-tetrazolium bromide (MTT) colorimetric assay[12], which is adapted to analyze cell proliferation and drug cytotoxicity.
(MATERIAL AND METHODS, page 9)
“12. Chen, Y.; Wang, Y.; Zhou, J.; Gao, X.; Qu, D.; Liu, C. Y., Study on the Mechanism of Intestinal Absorption of Epimedins A, B and C in the Caco-2 Cell Model. Molecules 2014, 19 (1), 686-698. doi:10.3390/molecules19010686”
[4. The authors must justify why they did not use a lower dose of 20 ug/ul of CAPB]
We appreciate reviewers’ comments. We assessed the transportation features of CAPB and CAPC from 10 to 40 μg/mL in caco-2 monolayers model, since caco-2 cell showed the loss of viability when the concentration of CAPB and CAPC was above 40 μg/mL. As results, we found that the Papp of CAPB and CAPC followed a linear increasing trend from 10 to 40μg/mL. However, we wanted to explore the intestinal permeability of CAPB and CAPC at higher concentration. So, we determined the range of 20-80μg/mL in SPIP model. To clarify these points, we have modified the DISCUSSION section as follows:
“However, we did not assess the transportation features of CAPB and CAPC at 80 μg/mL in the caco-2 monolayers model, as the drug showed cytotoxicity when the concentration of CAPB and CAPC was above 40 μg/mL. Because the Papp of CAPB and CAPC increased linearly from 10μg/mL to 40 μg/mL in caco-2 monolayer model. We wanted to further explore the intestinal permeability of compounds at higher concentration in SPIP model. ”
(DISCUSSION, page 8)
[5. The bibliographic collection is accurate, but some more recent references should be added. In particular:
- line 71, Authors should add “as observed for Cannabis sativa estracts” and insert “ Bonini SA et al. Cannabis sativa: A comprehensive ethnopharmacological review of a medicinal plant with a long history. J Ethnopharmacol. 2018”.]
We appreciate kind reminder. We have modified the sentences and added this conference as follows:
“Lysimachia capillipes Hemsl, a Chinese herb and medicinal plant, is widely used as a remedy for the treatment of colds and arthritis as observed for Cannabis sativa extracts [23-24].”
“24.Bonini, S. A.; Premoli, M.; Tambaro, S.; Kumar, A.; Maccarinelli, G.; Memo, M.; Mastinu, A., Cannabis sativa: A comprehensive ethnopharmacological review of a medicinal plant with a long history. J Ethnopharmacol 2018, 227, 300-315. doi:10.1016/j.jep.2018.09.004”
Reviewer 2 Report
Interesting, well performed experiment and well presented results and data.
The suggestions are addressed to the following:
Lines 90-92: This sentence is more for conclusion section. Please delete it.
Line 95: should be liquid….
Line 108: monolayers, please write it consistent through the text
Line 118 : a space should be inserted
Line 118 : should be written as 2×10-6
Line 177: caco-
Line 215: should be written as 2×10-6
Line 219: should be written as 0.2×10-4
Line 284: caco-
Line 306: should be written as 5×104
Line 319: monolayers
Lines 324 and 325: insert a space between the figure and unit
Line 380: caco-
Line 382: revised
Comments are also included in the pdf file of manuscript.

Author Response
XuZhang et al. MOLECULES/2019/ 464121
Dr. Derek J. McPhee
Editor-in-chief, Molecules
Reply to the Reviewers
Thank you very much for your interest in our manuscript entitled “In Vitro and In Situ Characterization of the Intestinal Absorption of Capilliposide B and Capilliposide C from Lysimachia capillipes Hemsl.” To aid in the re-review of this manuscript, we have included a point-by-point response to each comment. The reviewers’ comments are italicized and placed in square brackets. In addition, within the revised manuscript, we have used underlined text to highlight changes in response to the reviewers’ comments.
We appreciate the suggestions and comments by the reviewers. As a consequence of the valuable suggestions, we believe that our manuscript has been much improved.
[Reviewer 2: Comments for the Authors...]
[Questions and comments:]
[1. Interesting, well performed experiment and well presented results and data.
The suggestions are addressed to the following:
Lines 90-92: This sentence is more for conclusion section. Please delete it.
Line 95: should be liquid….
Line 108: monolayers, please write it consistent through the text
Line 118 : a space should be inserted
Line 118 : should be written as 2×10-6
Line 177: caco-
Line 215: should be written as 2×10-6
Line 219: should be written as 0.2×10-4
Line 284: caco-
Line 306: should be written as 5×104
Line 319: monolayers
Lines 324 and 325: insert a space between the figure and unit
Line 380: caco-
Line 382: revised]
Thank you for pointing out the errors in the manuscript. We examined the full text carefully and made a linguistic revision. To address the point raised, we have revised as follows:
Line 94: “by liquid chromatography-tandem mass spectrometry (LC-MSn).”
Line 107: “In our data, monolayer displayed a TEER values > 300 Ωcm2 and Papp values of Phenol <105cm/s after growing for 20 days.”
Line 117: “the Papp of CAPB and CAPC were all found to be less than 2×10-6 cm/s”
Line 213: “the Papp of CAPB and CAPC were both less than 2×10-6 cm/s”
Line 218: “Peff values of CAPB and CAPC were both much less than 0.2×10-4 cm/s”
Line 305: “uptake studies at 5×104 cells per insert.”
Line 323 and 324: “and 2.1 mL of HBSS on the basolateral (BL) side.”
Line 383: “The standard curve of CAPB and CAPC across caco-2 cell monolayer was detected by LC-MS.”
Line 385: “XuZhang wrote and substantively revised the manuscript.”